# Remote Controlled Nociceptive Threshold Testing Systems in Large Animals

**DOI:** 10.3390/ani10091556

**Published:** 2020-09-02

**Authors:** Polly Taylor

**Affiliations:** Taylor Monroe, Little Downham, Ely, Cambridgeshire CB6 2TY, UK; polly@taylormonroe.co.uk

**Keywords:** refinement, pain, nociceptive threshold, horse, cat, dog, sheep, camel

## Abstract

**Simple Summary:**

Measurement of the nociceptive threshold (NT) is widely used in the study of pain and its alleviation. This records the intensity of a stimulus that causes pain to the test subject. The end point of the test that indicates when the subject experiences pain, the NT, is a behavioural escape response. Detection of a reliable and repeatable response depends on the animal behaving normally throughout testing. Restraint and an unfamiliar environment may prevent the animal from displaying normal behaviour and impede acquisition of robust NTs. Remotely controlled testing enables NT data to be collected from unrestrained animals behaving normally. Development of a remote controlled system for measurement of thermal and mechanical NTs in a range of large animal species is described. Normal “baseline” thermal and mechanical NTs from untreated animals are reported. This information can be used to improve both the welfare of the animals under investigation and the quality of the data collected. Remote controlled systems are now in use worldwide in both the study of pain physiology and in developing new pharmaceutical and non-drug-based methods of pain relief.

**Abstract:**

Nociceptive threshold (NT) testing is widely used for the study of pain and its alleviation. The end point is a normal behavioural response, which may be affected by restraint or unfamiliar surroundings, leading to erroneous data. Remotely controlled thermal and mechanical NT testing systems were developed to allow free movement during testing and were evaluated in cats, dogs, sheep, horses and camels. Thermal threshold (TT) testing incorporated a heater and temperature sensor held against the animal’s shaved skin. Mechanical threshold (MT) testing incorporated a pneumatic actuator attached to a limb containing a 1–2 mm radiused pin pushed against the skin. Both stimuli were driven from battery powered control units attached on the animal’s back, controlled remotely via infra-red radiation from a handheld component. Threshold reading was held automatically and displayed digitally on the unit. The system was failsafe with a safety cut-out at a preset temperature or force as appropriate. The animals accepted the equipment and behaved normally in their home environment, enabling recording of reproducible TT (38.5–49.8 °C) and MT (2.7–10.1 N); precise values depended on the species, the individual and the stimulus characteristics. Remote controlled NT threshold testing appears to be a viable refinement for pain research.

## 1. Introduction

Nociceptive threshold (NT) testing is widely used in studies of pain, analgesia, hyperalgesia and allodynia. Such testing records the intensity of a stimulus that causes pain to the test subject. Ethical, humane and repeatable methods for applying the stimuli and evaluating the responses are essential for producing robust data and maximizing animal welfare. A ramped noxious stimulus, usually thermal, mechanical or electrical, is applied, and the intensity of the stimulus at which a clear aversive response occurs is taken as the nociceptive threshold [1]. The threshold is expected to increase after analgesic treatment and to decrease with hyperalgesia and allodynia. Nociceptive threshold testing is very widely used for translational studies in rodents, both for drug development as well as for fundamental study of the physiology of pain. This is exemplified in, for example, Gugliandolo and colleagues’ investigation into neuropathic pain in mice [2]. They used latency of the response to a thermal stimulus applied to the underside of the foot (Hargreaves test) and the withdrawal threshold for mechanical pressure on the plantar area of the foot (electronic von Frey filament test) to evaluate the hypersensitivity induced by sciatic nerve crush and its prevention by the treatment under investigation. Although rodents are not physically restrained for these tests, they must be contained in a special cage and be in close proximity to the tester, which may affect their behaviour [3]. Other tests such as the tail flick [1] and the Randall-Sellito method [4] used in rodents may involve actual physical restraint with even greater impact on behaviour. Larger animals are used less commonly for translational studies but their inclusion is essential for investigation into species-specific physiology and pharmacodynamics. The greater size of these species enables the equipment supplying the stimulus to be attached to the animal, for which rodents are too small. Remote control of this equipment without restraint or close proximity of the tester allows the animal to display natural behaviour.

There are a number of essential criteria for NT testing [5]. The stimulus must be repeatable over a few hours and reproducible over the weeks or months of a longitudinal study. The stimuli must not cause tissue damage, both for animal welfare and also so that the stimulated site is not sensitized to subsequent tests. It is essential that the stimulus is stopped as soon as the animal responds, and that there is a cut-out to prevent tissue damage if no response occurs. The end point of the test that indicates when the subject experiences pain, the NT, is a behavioural escape response and must be clearly identified. For instance, this includes stamping the foot or lifting the leg when a limb site is used. Turning to nose or bite at, and bending or jumping away from a site on the body is also seen. Detection of this response depends on the animal behaving normally throughout testing. Restraint and an unfamiliar environment may prevent the animal from displaying normal behaviour and impede acquisition of robust NTs. A relaxed, undisturbed animal ensures that responses are not suppressed by fear or distraction. It is often a requirement of studies investigating the effects of both analgesics (increased NT) and increased sensitivity (decreased NT) that the same animal can be tested at intervals over a period of at least 24 h. Testing over several days may be needed for investigation of, for instance, inflammatory pain and its treatment.

Thermal and mechanical testing has been used for many years in large (non-rodent) animals, particularly for analgesic drug development in the target species [6,7]. A thermal system designed specifically for cats was reported in 2002 [8] and has subsequently been widely used for analgesic drug evaluation in this species [9,10]. The thermal probe applied to the body was connected to the control unit by a cable. The cat was able to move around an individual cage while attached, but the cable was vulnerable to damage from the cat and caused some restriction of movement. The inherent disadvantages of a wired system were underlined by experience of its use in the cat. All the behavioural criteria described above apply to this species, but its unique temperament and desire to chase and bite the cables made a wireless control system highly desirable.

In horses and farm animals, the need for a cable connection generally led to the animal being restrained in stocks. Mechanical NT testing has also commonly been carried out in these larger species by exerting noxious pressure from a blunt ended pin pushed onto the surface of a limb [11,12,13]. These systems have all been driven by a variously sized and often noisy control box, again necessitating restraint in stocks. The animal is close to potentially fear-inducing or distracting machinery and human contact, which both prevent normal behaviour and limit the duration of a continuous study.

The potential advantages of testing unrestrained animals with a small and silent system were apparent. A wireless, remote controlled thermal NT testing system was first developed for cats and subsequently extended to larger animals, and then followed by a mechanical system for larger animals. Remote control of NT testing presents an additional challenge to purely telemetric recording as the power to drive the noxious stimulus must also be remotely controlled and attached to the animal.

This report describes the development and early validation of remote controlled systems for thermal and mechanical NT testing in a number of large (non-rodent) animal species. The data presented were collected during preparation for subsequent research protocols from animals tested in Australia, Brazil, Germany, Norway, Saudi Arabia, Switzerland, the UK and the USA under the authorization of the appropriate animal use ethical committee for each research establishment. Some of the data were presented at the World Congress of Veterinary Anaesthesia in 2006 [14] and to the Association of Veterinary Anaesthetists in 2008 [15].

## 2. Materials and Methods

### 2.1. Animals

All animals were housed, handled and fed according to the local institutional guidelines. All were confirmed as healthy from a clinical examination conducted during acclimatization. According to the species and their familiarity with the handling procedures, all animals were acclimatized to the testing environment for at least 30 min up to several days. Again according to the species, they were accustomed to wearing equipment for at least 30 min before testing. Naïve research animals wore dummy equipment for a few hours a day for several days before testing.

Cats: Group DEV was studied during the original development of the remote system. This group comprised 24 purpose-bred neutered (17 f, 7 m) domestic short haired (DSH) cats aged 1–4 years, weighing approximately 3–6 kg. They were group-housed and brought in pairs to the testing environment at least 2 h before any testing. Testing took place with the cats roaming freely in adjacent large wire-sided cages. Group CNSW comprised 2 adult neutered DSH cats housed at night and ranging freely outside during the day (6–8 kg). They were brought to the testing area at least 2 h before testing and were studied roaming freely in a large wire-sided cage. Group CWS comprised 2 young adult neutered DSH cats kept as household pets (3–4 kg). They were brought to the testing area at least 2 h before testing and were studied roaming freely in the room.

Dogs: Three groups of dogs were studied. Group DNSW comprised two mixed breed adult neutered male dogs (weighing 24 and 28 kg bodyweight). Group COL included 3 male and 3 female 8–9-month-old Walker hounds weighing 20–24 kg. Group WS comprised a Labrador, a spaniel and a mixed breed (15–25 kg). Groups DNSW and COL were kennel-housed in pairs or groups with lead and free range exercise daily. Group DWS dogs were kept as domestic pets. All dogs were brought to the study area at least two hours before any testing and studied whilst freely moving in large open-top wire cages or in the entire room.

Horses: Six groups of horses were studied. Group HPEN comprised four adult standardbreds, two mares and two geldings weighing 450–550 kg. They were turned out to pasture at night and stabled during the day. Testing took place in their own stall. Group HBR comprised 10 mixed breed young adult gelding horses weighing 268–460 kg. The horses were turned out on pasture at night and housed in part-covered yards for the duration of any testing schedule. They were brought to the testing stall and allowed at least 30 min acclimatization with the equipment in place before any testing. They were lightly restrained with a headcollar and long rope. Group HWS comprised two young Arab mares weighing 450 kg. They were kept at pasture and brought to the large open testing barn at least one hour before testing. They were also lightly restrained with a headcollar and long rope. Group HNSW comprised one adult Thoroughbred gelding weighing 475 kg. He was kept at pasture and allowed 30 min acclimatization before any testing, lightly restrained with a headcollar and rope in a small paddock. Group HNO comprised two adult Norwegian Trotter mares (450–500 kg). They were stabled throughout the study, and testing took place with them freely moving in their own stall. Group HHAN comprised seven adult warmblood horses weighing 650–700 kg, three geldings and four mares. They were also stabled throughout the study, and testing took place with them freely moving in their own stall. None of the groups was fed during testing.

Sheep: Two groups of sheep were studied. Group SWA comprised three calm, pregnant Merino sheep during preparation for a larger project investigating post-operative hyperalgesia. They were housed and tested in large, raised, wire-sided open-top pens in full view of their companions. They were allowed at least 30 min with the equipment in place before any testing. Group SZU comprised eight young adult Swiss Alpine sheep (castrated males and females, around 50 kg bodyweight). They were group-housed in large wire-sided open-top pens; testing was performed in a smaller area partitioned off with wire-sided hurdles, without separating the subject from its companions. At least 30 min acclimatization was allowed with the equipment in place before any testing started.

Camels: Three young adult dromedary camels were studied (group DCSA). One male and two females, weighing 6–700 kg, were included. They were housed in open yards but brought into individual horse stalls for the duration of the study. They were acclimatized to the testing environment for two days before any testing, which took place in their individual stalls.

### 2.2. Thermal system

The first remote controlled thermal NT testing system was aimed specifically for use in cats to replace the wired system developed in 2002 [8]. The remote system incorporated a similar thermal probe with the heating element adjusted to give the same heating rate (0.6 °C/s). The thermal probe comprised a heater and temperature sensor mounted together in heat conducting epoxy as a flat 10 × 10 mm probe weighing 5 gm [14] using battery power and an infra-red (IR) link (Maplins, UK). The probe was held against the cat’s shaved thorax with an elasticated band. Constant and repeatable pressure against the skin was maintained by inflating a modified blood pressure bladder behind the probe. At each test, the heater was activated, and when the cat responded by twitching the skin, jumping, turning, flinching or occasionally vocalizing, the stimulus was terminated immediately and the threshold temperature recorded. A maximum cut-off temperature of 55 °C was set to minimize skin damage if the cat did not react at lower temperatures.

For remote control, the equipment was miniaturized so that the circuit board and batteries could be carried by the cat on a 50 cm wide back pack positioned on the dorsal thorax (Topcat Metrology Ltd. WTT1) (Figure 1). The back pack was held onto the cat using an elasticated band, and the circuit board and batteries were secured using Velcro^®^, with vulnerable cables under a soft, flexible cover. The underside of the band held the bladder and temperature probe against the thorax. The pressure transducer for control of the bladder pressure was attached to the circuit board, and a window between 30 and 70 mmHg was indicated by illumination of red and green LEDs on the circuit board. Tubing between bladder, transducer and a non-return valve (used for manual bladder inflation with a 20 mL syringe) was housed under the soft covering. The sensor output was displayed on a 3-digit display with peak hold, also fixed onto the band. The whole system weighed 320 g (Figure 2). Heating was controlled by IR signal operated manually and was activated only when the remote control handset button switch was depressed, so was failsafe if operator or IR contact was lost. For each test, the heater was switched on and held on by the operator who released the button, thereby stopping the heating, when the cat reacted. The display unit held the peak reading at the point the heater was switched off, and this temperature was recorded as the threshold. Peak hold was overridden by a second IR control via the handset in order to read skin temperature before the start of each test. Mirrors mounted on the walls of the cage allowed the display to be read whichever way the cat was facing.

The remote controlled system was also used with dogs. It was intended to replace the wired system previously reported [16]. The feline equipment was modified only by using a longer thoracic strap to secure the back pack on the larger species. Heating rates of 0.6–0.8 °C/s were used, and the cut-out temperature was 55–60 °C. Testing on the thorax produced similar responses, including skin twitching, turning to the probe site, biting at the band, flinching and moving away, but rarely vocalization. Testing on a leg site was also used as described below for larger animals.

For larger animals, the electronics and batteries were mounted in a shaped box on the animal’s back, held in place by a larger version of the band used in cats and dogs (Topcat Metrology Ltd. WTT2) (Figure 3). Two digital temperature displays were used, mounted on each side of the control box. Two IR receivers were also used, mounted on each side. These modifications enabled IR control in the much larger accommodation space needed for larger animals and for the display to be read from outside the stall whichever way the animal was facing. In camels, a single cuboid box mounted on the side of the hump was adopted as the top of the hump was too high and rounded for the shaped box (Figure 4). A further development was to place the transmitter high on the wall of the stall, wired to the handset, and the receiver mounted on an aerial attached to the animal’s back (Figure 5a,b). A similar thermal probe as in cats was used, with a heating rate of 0.8 °C/s. Constant contact between probe and skin was assured with the same pressurized bladder. The temperature probe was positioned either on the thorax, under the control unit, or it was placed on the dorsal metacarpus or metatarsus using a smaller elasticated band and connected to the control box by a longer ribbon cable. On dogs, sheep, small horses and camels, the probe was secured on a limb without a pressure bladder; consistent contact was assured by careful adjustment of the elastic strap (Figure 6). A limb site was not tested on cats. The response to thoracic stimuli was a skin twitch, turning to the site, bending away from the site or becoming agitated. Limb stimuli evoked stamping, a snatched leg lift or nosing at the site. Cut-off was set to 55–60 °C.

A modified probe [17] was incorporated into the system for all species from 2013. The style of heating probe used in each group is indicated in Table 1. Heating rates of 0.6 °C/s were used in cats and dogs, 0.8 °C/s in horses and sheep, and around 2 °C/sec in camels.

#### Thermal Testing Schedule (see Table 1)

Baseline thermal nociceptive thresholds (TTs) were collected from conscious animals who had not received any medication. The TT for each animal was recorded as the mean of at least 3 tests recorded at 10 min intervals. When first applied, familiarization and training for both tester and subject comprised a series of up to 7 tests at no less than 10 min intervals in order to establish the reaction of the individual subject. This was repeated several hours later and usually again over the following few days to allow complete familiarization. Once training was completed, baseline TT was taken as the mean of 3–5 tests within 10%. Tests were conducted at ambient temperatures of around 21 °C in dogs and cats but ranging from 11 to 30 °C in the larger species.

During development in cats, 12 of group CDEV (CDEV1) were tested with both the original wired and the remote controlled systems in order to confirm that the remote system produced stimuli and TTs similar to the original. These cats had been previously familiarized with the original wired system. The remaining 12 cats were tested with only the new system (Group CDEV2). Four DEV1 cats (DEV1a) were tested without any drug treatment with both systems on separate occasions either on consecutive days or with not more than 12 months between testing days. Two cats were tested with the wired system first and two with the wireless first. Each cat was tested 5–13 times at 15 min intervals with each system. A further four CDEV1 cats (DEV1b) were also treated with opioids to examine the performance of the wireless system when thermal thresholds were raised above normal. Two cats received intramuscular (IM) butorphanol 0.4 mg/kg and two received sub-lingual (SL) buprenorphine 20 µg/kg. Three tests at 15 min intervals were made before treatment, and at intervals not less than 15 min post treatment for up to 24 h. The mean of the pretreatment tests was taken as the baseline TT for that individual (see Table 2).

Historical data from CDEV1 cats using the wired system [6] and CDEV2 cats using the wireless system [18] were compared before, during and after treatment with buprenorphine (20–80 µg/kg). Three to five tests were performed before treatment, and at intervals not less than 30 min for 24 h after treatment.

Thoracic baseline TTs were collected from cats in groups CNSW and CWS and from dogs in DNSW, DCOL and DWS. The CNSW group cats and the DNSW dogs each received 1.0 mg/kg methadone intramuscularly (IM) after baseline TT had been recorded. The TT was then recorded at 15 min intervals until it returned to baseline.

Baseline TTs were collected from horses in groups HPEN (thorax and legs), HBR (thorax and leg), HWS (thorax and leg), HNSW (thorax), HNO (thorax) and three from HHAN (thorax). Horses in groups HBR (*n* = 5), HNSW, HNO, HWS and HPEN were treated with intravenous (IV) 0.2 mg/kg methadone, 0.15 mg/kg methadone and 0.25 mg/kg xylazine, 0.2 mg/kg methadone, 0.1 mg/kg butorphanol and 0.03 mg/kg acepromazine or 0.5 mg/kg xylazine and 0.025 mg/kg butorphanol, respectively, after baseline TT had been recorded, and NTs were measured at 15–30 min intervals for 3 h and on the following day, 16–18 h later.

In sheep, the responses to thoracic stimuli were difficult to detect and this site was abandoned. Baseline limb TTs were collected in groups SWA (metatarsal site) and SZU (stifle and metatarsal area). The thoracic site was not attempted in camels, and only baseline leg TTs were recorded.

### 2.3. Mechanical System

The mechanical stimulus was produced by a single blunt ended pin driven onto the surface of the skin by a pneumatic actuator [19] positioned on the dorsal surface of the metacarpus, midway between the carpus and the metacarpo-phalangeal joint (Figure 7). Increasing pressure in the actuator drives the pin. The system is calibrated to give force (in N) as the stimulus intensity. The responses to stimulation of the limb are similar to those seen with thermal stimulation at the same site.

The remote controlled device (Topcat Metrology Ltd. WMT2) was first developed in horses to replace the wired system by modifying the manually operated system previously described [20,21,22], where pressure was produced in a syringe compressed by hand. Indicator lights were used to keep the force rise rate within a predefined window. The system was silent and allowed rapid removal of the stimulus at threshold via a vent valve. The remote controlled system was operated via IR signal from a handset activated by the assessor in the same way as for the thermal system. The modification to include the aerial on the animal’s back was also used for the mechanical system. The shaped unit was positioned on the horse’s back and secured with Velcro^®^ to a fly-sheet or surcingle. A pressure reservoir, recharged between tests, was mounted alongside the control unit (Figure 8). This supplied a miniature, silent, solenoid metering valve (Kinesis, UK) to increase the cuff pressure at a predetermined rate. The valve was controlled via a feedback circuit from a pressure transducer in the force actuator. The final vent valve configuration was “normally shut”, but opened once a second when in standby mode. This required less power, therefore improving battery life. It also ensured that the subject was familiarized with the regular click of valve operation, which although insulated to a low volume, could act as an audible clue if only present during the test. At the start of the test, the click simply changed from the vent valve to the pressure valve.

Threshold reading was held automatically and displayed on a digital display on both sides of the control unit. The system was failsafe with a safety cut-out at a preset force and a pressure relief valve with electronic interlock to remove the stimulus if the remote signal was lost. The system was validated by mounting the probe onto a force transducer (Kenwood, UK) and recording force rate rise (FRR) during calibration. Human thresholds were measured at 5 min intervals on the dorsal metatarsals of two of the researchers (PT, MD) to evaluate an initial pressure in the supply reservoir of 32, 43 and 49 kPa (240, 320 and 370 mmHg).

The mechanical system was too large for use in cats and dogs and was not tested in sheep, although at least those of over 40 kg would be suitable. The control box for camels was a cuboid design fitted to the side of the hump in the same way as the thermal system.

#### Mechanical Testing Schedule 

Baseline mechanical nociceptive thresholds (MTs) were collected from conscious animals who had not received any medication. The MT for each animal was recorded as the mean of at least three tests recorded at five minute intervals. When first applied, familiarization and training for both tester and subject comprised a series of up to 10 tests at no less than 5 min intervals in order to establish the reaction of the individual subject. This was repeated several hours later and usually again over the following few days to allow complete familiarization. Once training was completed, baseline MT was taken as the mean of 3–5 tests within 10%.

Baseline data were collected from horses in groups HPEN, HBR and four from HHAN (see Table 3). Horses in group HPEN were treated with 0.5 mg/kg xylazine and 0.025 mg/kg butorphanol IV after baseline MT had been recorded, and NTs were measured at 10–15 min intervals for one hour and on the following day, 20 h later. A set of baseline data was also collected from the four horses in group HHAN, each test within 2 min of the same time points tested with the wired system previously described [21] and attached to the same actuator. In this case, the tester stood close to the horse and manually operated a syringe to generate the stimulus.

Baseline data were collected from the three DCSA camels using a 0.5 mm probe tip at a mid-metacarpal site (see Table 3). The actuator was positioned on the lateral aspect to prevent it from being dislodged when the camel lay down. Data were collected at 10 min intervals with a force rise of 2 N/s.

### 2.4. Data Analysis

Descriptive data are presented, including range and mean ± SD as appropriate. An unpaired *t*-test on the excursions (ΔT °C = threshold temperature–skin temperature) was used to compare the data collected with wired and wireless systems in cats and horses. Friedman’s test was used to compare baseline data with NT after opioid treatment as the data are not normally distributed due to “capping” of the peak TT by the cut-off (GraphPad Prism v8). *p* < 0.05 was regarded as significant.

## 3. Results

### 3.1. Thermal System

#### 3.1.1. Evaluation of Remote Control (Table 1)

The IR control was successful in all the species, and there were few incidents where contact was lost. The signal was used either by pointing the transmitter directly at the control unit on the animal or by intentionally reflecting the signal off a reflective surface. The only materials that proved inadequate for reflection were rough high wooden roofing in a barn or horse bedding on the ground. Inclusion of the aerial on the animal and a fixed transmitter position prevented loss of contact through human error in misdirecting the signal and facilitated separation of the tester from the animal. Low ambient temperature did not affect the unit’s function, except that more power (i.e., a fully charged battery) was required to heat to threshold temperature from a lower starting point.

#### 3.1.2. Comparison with Wired System

In the four DEV1a cats, mean ± SD baseline with the remote system was 43.6 ± 2.1 °C and with the wired system 42.0 ± 1.7 °C. Excursions (TT – skin temperature) in four cats (DEV1a) were 5.0 ± 1.1, 2.8 ± 0.9, 3.0 ± 0.5, 4.0 ± 2.1 °C (remote) and 4.4 ± 1.3, 3.4 ± 1.3, 5.3 ± 1.3, 6.1 ± 1.0 °C (wired). Excursions measured with both systems were not statistically different. Peak excursions in two cats were >13.1 °C (reached cut-out) 10–105 min after butorphanol, similar to previous data from opioid-treated cats using the original system [23,24].

In group DEV1a cats (*n* = 4), the skin temperature was always higher when recorded by the remote system compared with the wired (wired 36.8 ± 0.6 and remote 39.4 ± 1.5, *p* < 0.001). Remote TT was higher than wired in only one cat, and the mean difference was small although significantly different (wired 42.0 ± 1.7 and remote 43.6 ± 2.1, *p* < 0.01). However, the excursions recorded with the remote system were smaller than with the wired (wired 5.2 ± 1.4 and remote 4.3 ± 2.0, *p* < 0.05). The TT in all Group DEV1b (*n* = 4) cats increased after opioid treatment. Peak excursions ranged between 16.1 and 18.2 °C, which is similar to published data (Table 2).

**Table 2 animals-10-01556-t002:** Thermal thresholds (TTs) in group DEV1b cats. Peak excursions (skin–TT difference) after treatment with intramuscular (IM) butorphanol 0.4 mg/kg or sub-lingual (SL) buprenorphine 20 µg/kg. Published approximate mean peak delta T data after buprenorphine and butorphanol included for comparison.

	Maximum Excursion °C	
Cat/reference	*Butorphanol*	*Buprenorphine*
5	18.1	
6	16.4	
7		18.2
8		16.1
Lascelles and Robertson [23]	18	
Robertson et al. [24]		17

**Table 3 animals-10-01556-t003:** Mechanical nociceptive thresholds (MTs) (mean ± SD and range) using a limb-mounted actuator under infra-red remote control during training in horses and camels.

Species, Group	MT (N)	Pin Diameter	Max MT After Opioid (N)
*Horse*			
HPEN (*n* = 4)	10.3 ± 4.6 (3.0–15.0)	3 pin	20.3
HBR (*n* = 10) (leg)	5.6 ± 2.3 (2.7–10.1)	1 mm	n/a
HHAN (*n* = 4)	6.3 ± 2.2 (4.0–9.9)	1 mm	n/a
All horses (1 mm pin)	5.9 ± 2.2		
*Camel*			
DCSA (*n* = 3)	13.8 ± 2.3 (11.2–15.5)	0.5 mm	n/a

The historical data from CDEV1 cats (*n* = 12) using the wired system [6] and CDEV2 cats (*n* = 12) using the wireless system [18] showed that mean skin temperature in CDEV1 cats was always 1–2 °C lower than in CDEV2 cats. The TT in both groups increased significantly after buprenorphine treatment, remaining higher than pretreatment from 60 to 240 min in CDEV1 and from 30 to 300 min in CDEV2. The TTs were similar except 1–2 h after treatment, when CDEV2 TTs were higher.

#### 3.1.3. Cats

The cats were undisturbed by the equipment and behaved normally whilst wearing it for several hours. The only impediment in those freely ranging in a room was inability to enter a space only as wide as the cat, when the back pack was caught and slid back. This did not occur in animals loose in a large cage. Cats treated with opioids tended to roll over; this sometimes caused the back pack to slip and require adjustment before a reading could be made. Response at TT was a clear skin flick or turn towards the heated site. Occasionally the cat would jump forwards at TT. Vocalization was rare. Group CNSW baselines ranged from 43.6 to 46.4 °C, reached cut-out (55 °C) 30 min after IM methadone (1 mg/kg) and remained higher than baseline for 3 h. Group CWS cats’ baselines were within the same range (see Figure 9a,b).

#### 3.1.4. Dogs

The dogs rapidly became accustomed to the equipment and behaved normally whilst wearing it for several hours. Younger, more active dogs required more acclimatization than calmer animals. They tended to scratch at the back pack with a hindleg when it was first put on. Clear responses included a skin flick, turning to, scratching or biting at the site and jumping forwards. Vocalization was very rare. Baseline TT in group DNSW dogs ranged from 42.7 to 44.6 °C, reached cut- out (55 °C) 10 min after IM methadone (1 mg/kg) and remained higher than baseline for 3 h. In group COL, mean ± SD baseline skin temperature was 37.7 ± 0.3 and TT 47.9 ± 1.4 °C (see Figure 9a,b).

#### 3.1.5. Horses

All the horses accepted the equipment without any reaction. Occasionally it was pushed posteriorly if a horse rubbed on a wall or fence, but repositioning was rarely required. Responses at TT on the leg were a clear leg lift or stamp, or occasionally the horse would nose or even bite at the site. Responses to the thoracic site were more difficult to detect as a skin flick could be obscured by the more bulky testing equipment used in large animals and the horses appeared less responsive at this site. However, a skin flick, a bend of the body away from the testing site or turning to look at the site were also commonly seen. Skin temperature was lower on the leg than the thorax and was more dependent on ambient temperature. The TT was higher on the thorax than on the leg. Group HBR skin temperatures were 36.7 ± 0.4 (thorax) and 32.7 ± 1.0 (leg), and TT 54.8 ± 1.8 (thorax) and 46.9 ± 3.2 °C (leg). Group HPEN skin temperatures were 36.7 ± 0.4 (thorax) and 32.7 ± 1.0 (leg), and TT 54.8 ± 1.8 (thorax) and 46.9 ± 3.2 °C (leg). Leg skin temperature and TT were significantly lower than on the thorax (*p* < 0.001). Skin temperature and TT ranges in the smaller groups (HWS, HNO, HWS) are shown in Table 1. The TT in horses given analgesic treatment (groups HBR, HNSW, HNO, HWS and HPEN) increased above baseline between 10 min and 1–2 h. It reached cut-out (55 or 60 °C) in all except HPEN horses where the highest TT was 53 °C. All TTs had returned to baseline three hours after dosing and were within the same range the following day (see Figure 9a,b).

#### 3.1.6. Sheep

All the sheep accepted wearing the back pack with very little acclimation time. The calm SWA sheep tolerated the testing protocol without any problems, and the response at TT was a clear leg lift or stamp. Skin temperature and TT ranges are shown in Table 1. The young flighty Swiss mountain sheep were much more difficult to test as they appeared to adopt a prey species freeze when agitated or frightened by anything unfamiliar, particularly the presence of people. Many tests went to cut-out with no response detectable. When the tester was hidden outside the room responses were lower, in accordance with values reported in other sheep [25]. This is demonstrated by sheep 5201 with a leg skin temperature of 34.3 °C: with a tester standing in the same room, the temperature consistently reached cut-out with no response. When the tester was obscured from the sheep outside at a window, and outside the pen, TT was 53.7 °C (see Figure 9a).

#### 3.1.7. Camels

The camels willingly accepted the equipment on their backs. Shaving the skin to enable contact with the thermal probe caused more aggravation than wearing the limb bands once they were positioned. The response at threshold was a clear leg lift or stamp. However, the camels tolerated much higher temperatures than other species, and custom probes with a faster heating rate were developed for them. Even after reaching temperatures in excess of 100 °C, there were no skin lesions other than a slight indent under the footprint of the probe. No pain, swelling or abrasion of the skin was seen. Skin temperature in the DCSA group was 26 ± 1.6 and TT 95.0 ± 14.3 °C (see Figure 9a).

### 3.2. Mechanical System

#### 3.2.1. The System (Table 3)

The unit functioned as intended; the solenoid metered valve resulted in an FRR that did not deviate by more than 0.1 N regardless of the initial reservoir pressure. IR control was successful in both species tested, as for the thermal system. The continuous quiet click of the vent valve during standby and of the pressure valve during the test was ignored by all the subjects, confirming the benefit of the regular opening of the vent valve during standby between tests. If the signal contact was lost for any reason, however, the failsafe system vented the pressure in the actuator and that test was aborted. Incorporating the fixed transmitter and the aerial mounted on the animal’s back ensured that this was extremely rare, and there were few incidents where contact was lost. Human MT from five tests was 7.3 ± 0.3 N, consistent with data from previous devices [21], with a coefficient of variation of 3.6%. Cut-out was set at 20–25 N.

#### 3.2.2. Horses

The horses tolerated the equipment in the same way as the thermal unit, and both could be accommodated on the horse’s back at the same time. The response at MT was indistinguishable from the response to thermal stimulation at a similar site, namely a leg lift, stamp or turning to and nosing the actuator. Baseline MT in the HBR, HPEN and HHAN (*n* = 4) groups was 5.6 ± 2.3 N, 10.3 ± 4.6 N and 6.3 ± 2.2 N, respectively (See Figure 10a). In the HPEN group, MT increased above baseline for 30–60 min after the analgesic treatment. It returned to baseline by 75 min and remained at baseline (7.9 ± 2.5 N) 24 h later ((See Figure 10b). The MT measured simultaneously with the wired system (4.1 ± 1.3 N) in the four HHAN horses was significantly lower (*p* < 0.05) than when measured by remote control ((See Figure 10c).

#### 3.2.3. Camels

The camels tolerated the mechanical equipment as easily as the thermal, and both units were worn together for several hours at a time with no effect on normal behaviour. Responses to stimulation were a clear leg lift or stamp. The DCSA group baseline MT was 13.8 ± 2.3 (See Figure 10a).

## 4. Discussion

This report documents the development and evaluation of remote controlled systems for TT and MT testing in large animals. Remote control by IR of both thermal and mechanical systems proved reliable. The baseline data were consistent, and increased thresholds were detected after analgesic treatment. Although NT testing does not mimic the complexities of true clinical pain, enabling normal behaviour should bring the response to a drug closer to clinical reality than in a restrained animal and therefore provide information that more closely resembles how a drug will behave under clinical conditions. The adaptations required both for power to activate the stimulus and for the control circuits to be placed on the animal were successful. Thermal stimulation depended only on small electrical components producing heat, and the main requirement was sufficient miniaturization for the electronics and sufficient battery power to be carried on the back of an animal as small as the cat. The mechanical system, however, required generation of air pressure to drive the actuator pin, necessitating attachment of a gas reservoir and metered pressure control being fitted to the animal. Wired or handheld MT testing systems driving a pneumatic actuator can make use of manual compression of a syringe [21,22,26]. Both air reservoir (the syringe) and pressure control (e.g., following visual pressure indicators) are under operator control and not fixed to the animal. The system adopted for the remote control system incorporated solenoid metering valves and an air reservoir of at least 60 mL. This proved too large for dogs and cats, but suitable for larger animals.

Most of the criteria for NT testing outlined by Beecher [5] were met. The aspects concerning repeatability of the stimulus, minimal tissue damage, immediate cessation of the stimulus when the animal responds and automatic cut-off if there is no response are no different from wired or handheld devices [8,22]. Features to prevent tissue damage and to ensure stimulus repeatability have been addressed elsewhere [17]; this report concentrates on aspects that allow normal behaviour and a clear escape response at threshold.

All the species tolerated wearing the equipment well. The larger, herbivore animals often required no acclimation to this at all. Young, playful dogs and cats required more “dummy” sessions, but reliable data could be collected in all cases with time and handling appropriate to the species. Complete free roaming was clearly not possible as the IR system would be out of range and the animal must be within sight of the tester for its response to be seen. However, testing was entirely feasible within a room or large kennel for small animals and in a stall without the use of stocks for large animals. Absence of wires and cables was particularly valuable in cats and young, playful dogs.

In contrast to handheld units, remote NT testing limits the stimulus site to body parts where the probe or actuator can be reliably fixed to the animal’s body. Mechanical testing in particular is limited as the actuator must react against something to generate the force. A band around a limb provides the necessary configuration, but application to other parts of the body is more difficult. Although fixing the thermal probe on the thorax was both straightforward and suitable for cats and dogs, in the larger animals it was sometimes difficult to detect the end point, in part as the thorax was obscured by the bulkier equipment but also because the response itself was less clear. Independent investigations examined the response to NT at different sites, and both concluded that the thorax was less reliable and NTs were higher [27,28]. Large animals in stalls are more likely to be bothered by flies, resulting in confusing skin twitches over the body. Testing the limb is generally preferable, except when opioid analgesia is employed when an alternative site maybe better. Opioids often produce locomotor stimulation in otherwise pain-free healthy horses and obscure the leg lift or stamp response. In cold weather, a further disadvantage of the limb site for thermal testing was the low skin temperature of the limbs requiring more power to heat to threshold temperature. It is of note that skin temperature and TT in the cats were higher with the remote system than the wired. This was considered likely to be due to the larger bulk of the remote band providing more insulation than the narrow band without circuit boards used for the wired system. The original wired probes also included a substantial aluminium support that conducted heat away, probably contributing to lower indicated skin temperature. However, in spite of the higher temperatures recorded with the remote system, the excursions (TT – skin temperature) were similar. This has led to use of the excursions in many reports [29], at least for statistical analysis, as it removes one source of variation.

The precise NTs recorded are dependent on many factors including the site, the style of the stimulus probe, the environment and the breed or strain of the species under study. This makes results from studies in different laboratories difficult to compare. Remote testing may reduce the variation by allowing normal behaviour but it still remains extremely important that all the conditions are detailed, and strict consistency within one study is essential. A further central source of disparity within a study is individual variation. There is usually a range of skin temperature and both TT and MT even within a group of similar animals [30], and reducing all other sources of variation is necessary to produce useful data. It is important to describe the site, heating rate, probe style and the environment in order to understand the data fully.

It is well recognized that distraction alters the perception of pain [31] and, however caused, may affect recorded NT. Furthermore, the importance of enabling a natural escape response to indicate the NT has been emphasized [32]. Any distraction or, particularly in prey species, anything threatening or frightening will illicit abnormal behaviour, particularly freezing, and may prevent a natural escape response [33]. The potential for the tester to be unseen by the animal subject is of particular benefit in prey species unaccustomed to contact with humans. This was illustrated by the sheep who did not respond when the tester was in the same room. Our own unpublished observations with rabbits support this: there was no response to thermal stimulation unless the tester was hidden, when TTs similar to dogs and cats were recorded. The effect of close proximity of the tester was illustrated even in calm horses who were quite familiar with humans. The MTs in the HHAN horses were significantly lower when tested with the wired compared with the remote controlled system. This is presumably due to an element of anticipation and apprehension with close proximity, when there are visual and perhaps audible clues that the stimulus is coming. This effect is further illustrated in sows [22] where hand-held MT testing was compared with a limb-mounted actuator, albeit still connected by a light pressure line. Baseline MTs were 13–17 N with the handheld device and 18–23 N with the limb fixed actuator, incorporating the same probe tip.

This investigation did not address reproducibility of the NT in the same animals over days, weeks or more. Remote control would in theory probably foster stable baseline thresholds. Learning to anticipate an aversive experience is less likely if the animal’s behaviour is unaffected and no cues reinforce anticipation. Dogs undergoing MT testing with a handheld device at several sites at 10–15 s intervals had lower MTs when tested 10–14 days later [34]. It was concluded that the dogs had learned to respond to the same stimulus more quickly. This effect was probably exacerbated by the necessary restraint, the high frequency of the tests and the use of a large probe tip (1 cm diameter), which requires a larger force to produce the same pressure as the smaller probe tips used here. Pain is experienced when the stimulus intensity is sufficient to stimulate nociceptive nerve endings by pressure; the high forces required to generate sufficient pressure with a large probe surface area may squash the dog and be unpleasant even before they produce pain. Remote control does not address all these potential confounders; care with the frequency of repeated stimuli and the forces applied are equally important as lack of restraint and freedom of movement.

In spite of this experience with a handheld device [35], a number of studies using remote controlled NT testing have demonstrated good reproducibility. In dogs, TTs were shown not to change over daily testing for 3 days [36] and in cats over several months [6]. In horses, neither TT nor MT changed significantly over several weeks [27,37,38,39]. Pastern MT remained stable or even increased with familiarity over a few weeks, even when attached to the operator by a long light pressure line [26]. The ability of remote control to avoid visual cues and allow normal behavior presumably contributes to this.

Neither TT or MT remote control testing were evaluated in cattle, nor MT testing in sheep. In principle, these should both be suitable species for remote controlled systems, being large enough to carry the bigger control boxes on the back. Sheep tolerated the back-mounted remote thermal control system willingly. Actuators applied to limbs have been used in cattle standing in stocks [12,13], and it seems likely that the remote systems would have similar advantages over wired in the same way as in horses. Thermal threshold has been measured successfully in both neonatal and young foals [40].

Camels reacted to MT testing in a similar manner to cattle and horses, and the remote system functioned well in relatively recalcitrant animals once all the equipment was in place [41]. Thermal testing in camels was challenging: the high limb thresholds are presumably a result of adaptation to withstand desert and sandstorm temperatures of over 50 °C.

The IR remote controlled systems have now been used for formal investigations into pain and analgesics in cats [42,43], dogs [29], sheep [25] and horses [37,38,39,44].

## 5. Conclusions

Remote controlled thermal nociceptive threshold testing in cats, dogs, sheep, horses and camels and mechanical threshold testing in horses and camels was well tolerated. It allowed free movement and normal behaviour during testing and resulted in data consistent with wired systems. Remote controlled nociceptive threshold testing is a useful refinement for pain research as the animal can behave normally without restraint or proximity of the tester.

## Figures and Tables

**Figure 1 animals-10-01556-f001:**
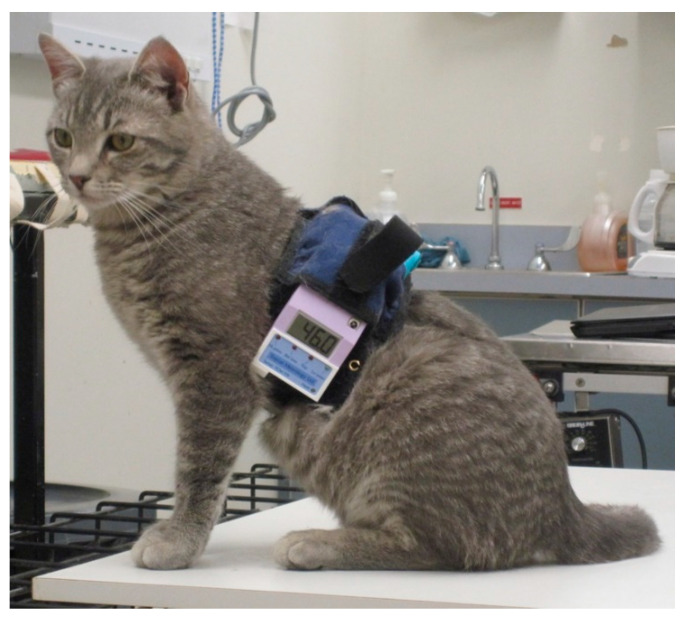
Remote controlled thermal unit worn by a small cat (3 kg) free roaming in a room.

**Figure 2 animals-10-01556-f002:**
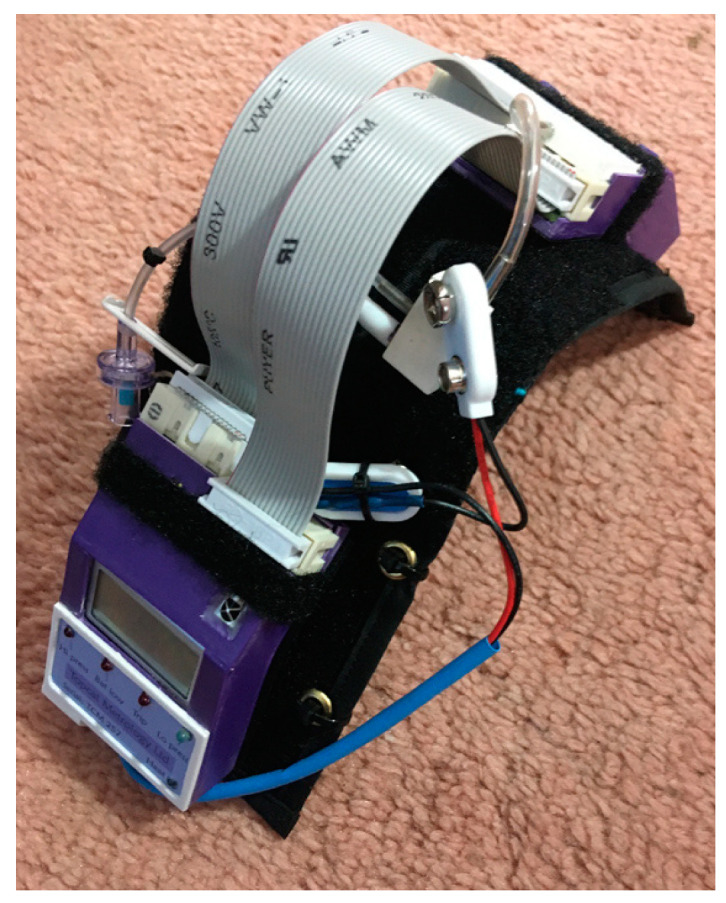
Remote controlled cat thermal band with cover removed.

**Figure 3 animals-10-01556-f003:**
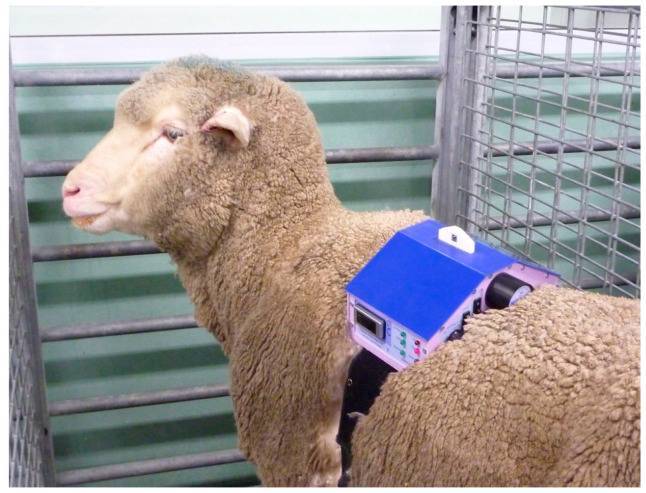
Remote controlled thermal unit on an unrestrained sheep in its pen.

**Figure 4 animals-10-01556-f004:**
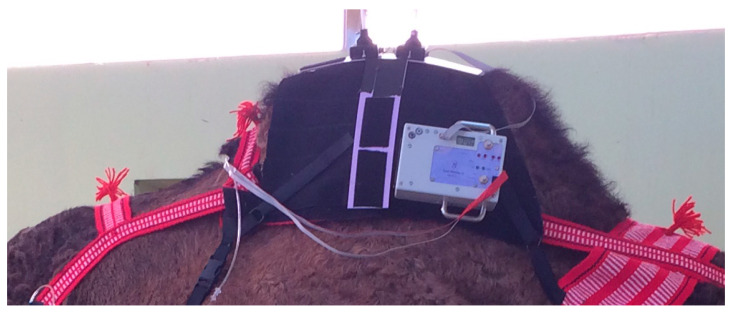
Cuboid thermal unit mounted on a camel’s hump.

**Figure 5 animals-10-01556-f005:**
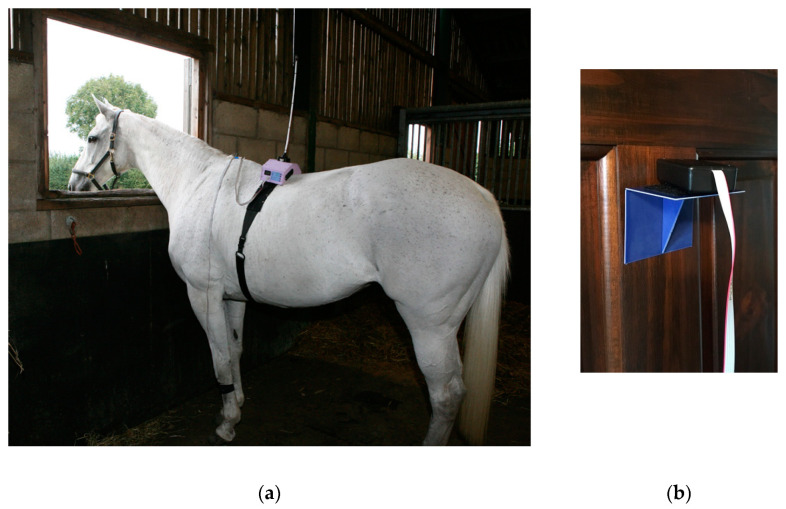
(**a**) Thermal unit with receiver mounted on aerial on a horse’s back. (**b**) Transmitter placed high on stable wall.

**Figure 6 animals-10-01556-f006:**
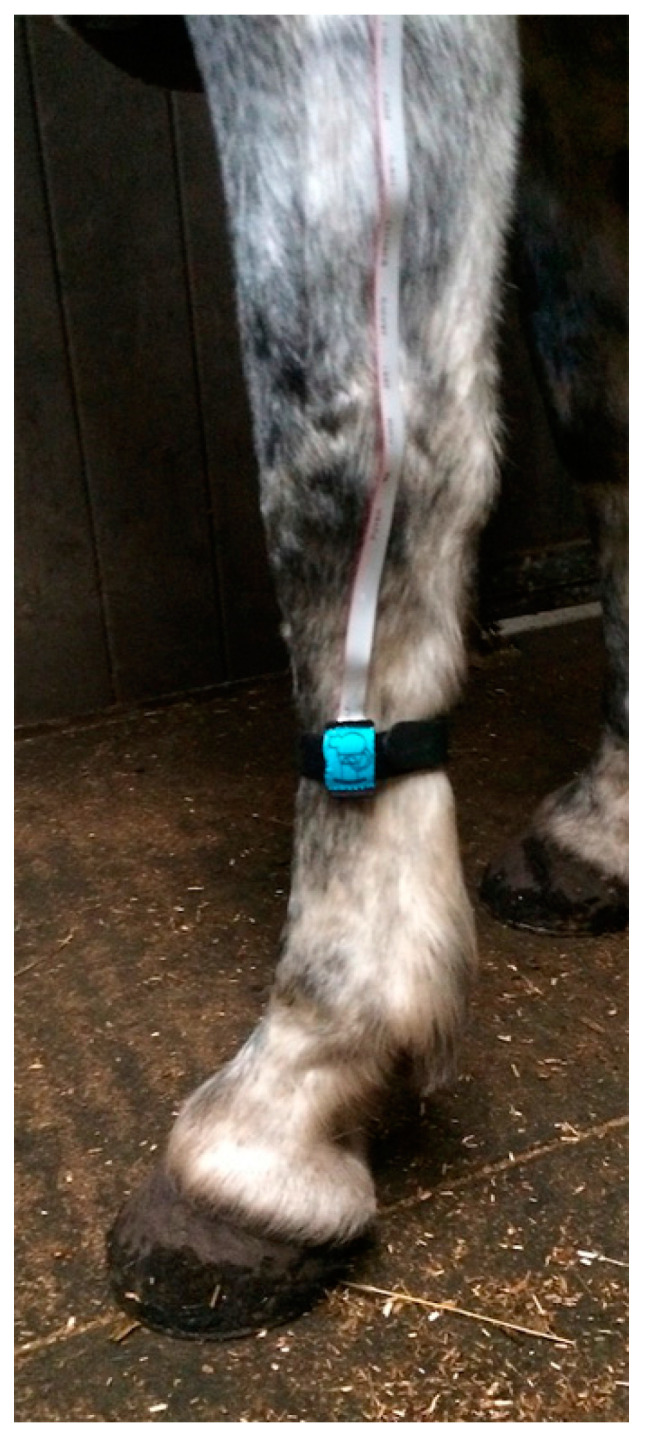
Thermal probe positioned on the limb with a carefully tensioned elastic and Velcro^®^ band.

**Figure 7 animals-10-01556-f007:**
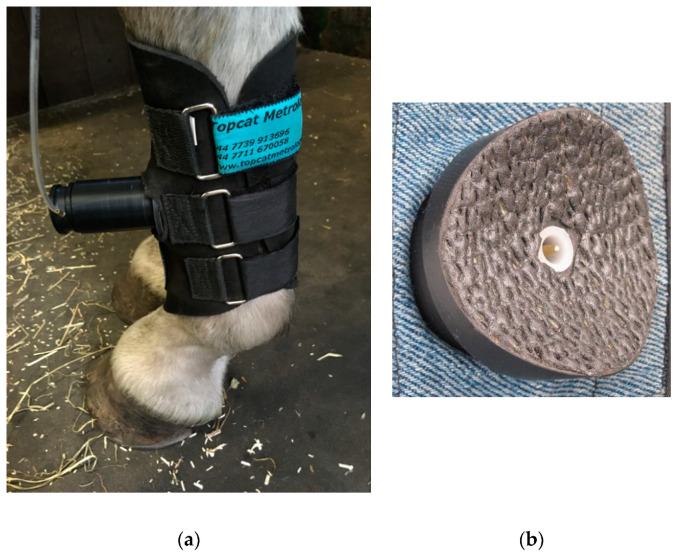
Pneumatic actuator: (**a**) secured with boot and carefully tensioned band on the forelimb of a horse standing free in its stall, (**b**) the leg side of the actuator showing probe tip in resting position.

**Figure 8 animals-10-01556-f008:**
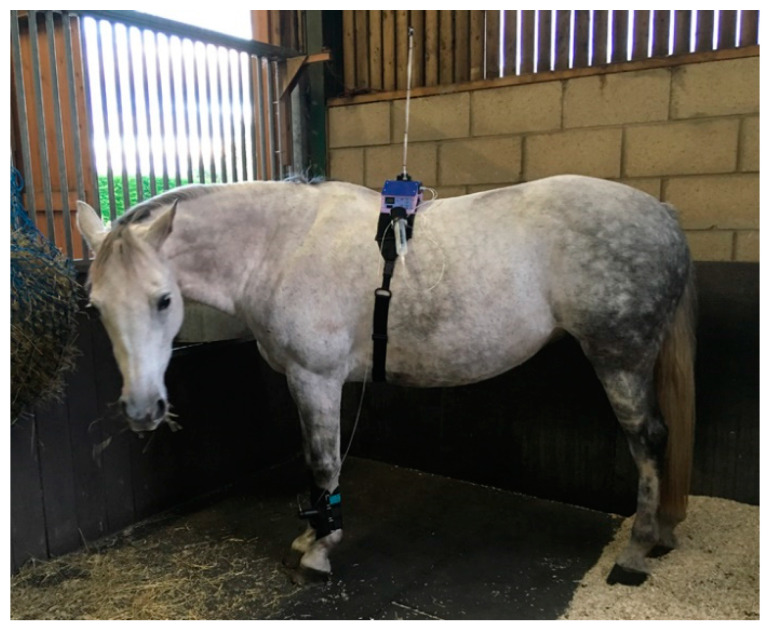
Horse with mechanical system in place: control unit and reservoir mounted on the back, with pressure line to the actuator on the foreleg.

**Figure 9 animals-10-01556-f009:**
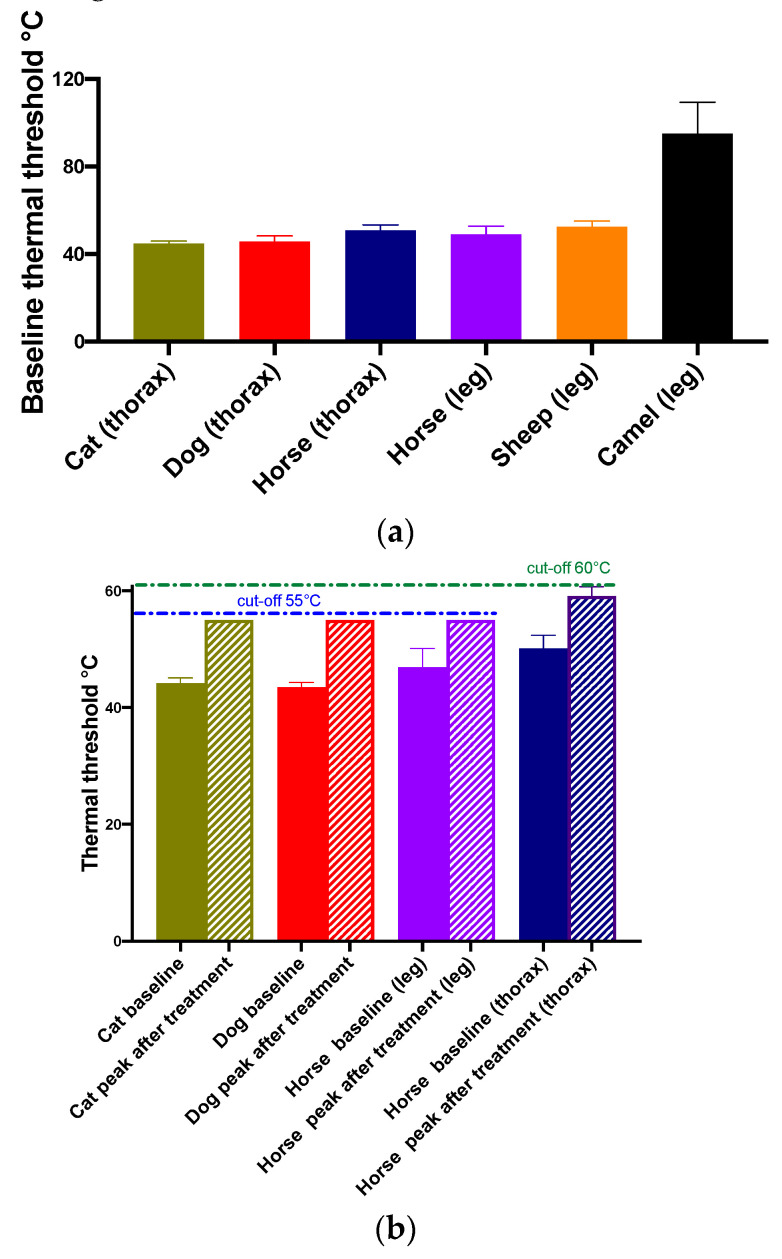
(**a**) Mean ± SD baseline (no treatment) thermal nociceptive thresholds (TTs) (°C) measured on the thorax in cats (*n* = 16), dogs (*n* = 8) and horses (*n* = 21) and on a leg in horses (*n* = 15), sheep (*n* = 11) and camels (*n* = 3). (**b**) Mean ± SD pre treatment and peak post analgesic treatment (see text for detail) TTs (°C) measured on the thorax in cats (*n* = 6), dogs (*n* = 2) and horses (*n* = 8) and on a forelimb in horses (*n* = 10). Thermal threshold reached cut-out (55° or 60 °C as shown) in all cats, dogs and in horses using the leg site. For the horse thoracic site, those reaching cut-out (*n* = 4) were deemed a TT of 60 °C for illustrative purposes.

**Figure 10 animals-10-01556-f010:**
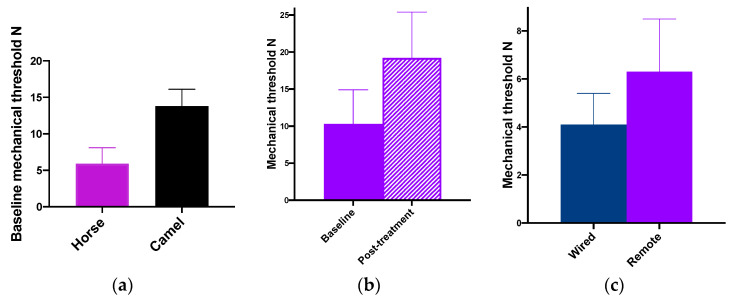
(**a**) Mean ± SD baseline (no treatment) mechanical nociceptive thresholds (MTs) (N) measured on a leg in horses (*n* = 14) and camels (*n* = 3). (**b**) Mean ± SD pre treatment and peak post analgesic treatment (see text for detail) MTs (N) measured on a foreleg in four horses using a 3-pin actuator. (**c**) Mean ± SD baseline (no treatment) MTs (N) measured on a foreleg of 4 horses with a wired system (close proximity and contact with the tester) and with the infra-red remote controlled system (horse unrestrained and tester outside the stall). Measurements with each system were made within less than 5 min.

**Table 1 animals-10-01556-t001:** Skin temperature (°C) (range) and thermal nociceptive threshold (°C) (TT) (mean ± SD (range)) recorded with a remote controlled system during training in cats, dogs, sheep, horses and camels.

Species, Group (site)	Skin Temp (°C)	TT (°C)	Probe Style	Max TT (°C) After Opioid	Cut-Out (c/o) Temp (°C)
*Cat*					
CDEV *n* = 12 (thorax)	38.6–39.6	43.2 ± 3.0 (41.2–45.8)	old	c/o	55
CNSW *n* = 2 (thorax)	36.2–37.1	45.1 ± 1.8 (43.6–48.1)	new	c/o	55
CWS *n* = 2 (thorax)	36.2–36.4	45.8 ± 0.1 (45.7–45.9)	new	n/a	60
All cats (thorax)		44.7 ± 1.1			
*Dog*					
DNSW *n* = 2 (thorax)	35.9–36.6	43.5 ± 0.8 (42.7–44.6)	new	c/o	55
DCOL *n* = 6 (thorax)	37.3–38.2	47.9 ± 1.4 (45.9–49.8)	new	n/a	60
DWS (*n* = 3) (leg)	30.4–33.0	42.7 ± 1.4 (38.4–5.3)	new	n/a	60
All dogs (thorax)		45.7 ± 2.5			
*Horse*					
HBR (*n* = 10) (thorax)	36.3–37.3	54.2 ± 2.0 (51.9–58.4)	new	n/a	55
HBR (*n* = 10) (leg)	31.5–34.0	46.9 ± 3.2 (41.0–51.3)	new	c/o	55
HWS (*n* = 2) (leg) (thorax)	35.7 34.8	54.4 52.7	new new	n/a c/o	60 60
HNSW (*n* = 1) (thorax)	36.2	50.9	new	c/o	60
HNO (*n* = 2) (thorax)	33.7–36.4	50.5 ± 3.4 (45.0–54.4)	new	c/o	60
HHAN (*n* =3) (thorax)	35.8–36.8	50.5 ± 3.2 (47.2–56.0)	old	n/a	60
HPEN (*n* = 4) (thorax)	31.1.–31.7	46.4 ± 1.9 (44.5–48.2)	old	n/a	60
HPEN (*n* = 4) (leg)	21.0–30.5	45.7 ± 1.7 (43.4–47.3)	old	56.2	60
All horses (leg)		49.0 ± 3.8			
All horses (thorax)		50.9 ± 2.4			
*Sheep*					
SWA (*n* = 3) (leg)	32.3–35.3	51.2 ± 1.9 (48.8–54.1)	new	n/a	60
SZU (*n* = 8) (leg, stifle)	30.5–34.9	53.7 ± 3.0 (48.9–c/o)	old	n/a	70
(leg, metatarsal)	18.0–26.2	55.0 ± 1.6 (53.0–c/o)	old	n/a	70
All sheep (leg)		52.4 ± 2.6			
*Camel*					
DCSA (*n* = 3) (leg)	24.2–27.2	95.0 ± 14.3 (74.9–106.0)	custom	n/a	130

Probe: old style [8], new style [17]. Cut-out (c/o): temperature set to cease heating automatically when no response is detected. Opioid treatment: butorphanol, methadone, buprenorphine or methadone/sedative combinations as detailed in the text.

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
