# Peer review of "Remote Controlled Nociceptive Threshold Testing Systems in Large Animals"

_animals, 2020, doi:10.3390/ani10091556_

Round 1

Reviewer 1 Report

Major:

The main problem of this manuscript is the NT was not measured alone only using analgetics, but the animals used in this study had addionaly other influences by their respective study protocol. This has to be  discussed in more detail.

minor:

Line

comment

34

and the individuum (animal)?

110

Where the animals clinically checked before entering the study?

135

The feeding of the horses took place in the stable or on the pasture? This might be a confounder for the results, because the horses are more willingly to go to the place where they are fed.

175

use abbreviation of SI-units

268

A table of all species and groups with their respective treatment would be appreciated and elucidate the latter results section tables

340

What statistical program was used?

362

difference of ca to c/o. Please explain at the end of each table all abbreviations.

495

You also report results on cats.

600

The influence of the different treatment beside NT should be discussed.

644

Name?

Reviewer 2 Report

The study describes an interesting topic however there are some points that should be improved:

  • line 87: each individual approval number for the animal study should be specified for each country mentioned.
  • In the introduction insert the following quote mentioning the methods used in rodents 10.1186 / s12974-018-1303-5
  • Authors should provide a photo of the inside of (the part that comes into contact with the animal skin) the Pneumatic actuator
  • also providing an additional cumulative graph for each species could better render the values shown in the table
  • How could the results presented in this study be reflected in clinical practice? authors should better discuss this in the discussion

Round 2

Reviewer 1 Report

Thank You to the authors to answer my questions profoundly, even when they explained me why they didn't change one item.

Author Response

Thank you for your comments.

Further changes have been made in response only to reviewer 2.

Reviewer 2 Report

- As the authors said the methodology described is only appropriate for research purposes. In this context the previous request to include details of the methods used in rodents is reasonable. In fact the methods described in the previously indicated reference describe the basis of the methodology described in this study. As the plantar test and the Von Frey test (thermal and mechanical stimuli) are widely recognized as important tests for the nociceptive study in various experimental models and species. The author in the introduction should better explain the methodological basis of the use of mechanical and thermal stimuli.

- The author should justify the subdivision into sub-groups that has been made for each species. es for horse HBR (n = 10) (thorax) HBR (n = 10) (leg) HWS (n = 2) (leg)(thorax) HNSW (n = 1) (thorax) HNO (n = 2) (thorax). A cumulative mean for each species should be shown (in table or graph) as it would give more information on a larger sample size. The results thus obtained would give more information (average and SD or SEM) on the reproducibility of the methodology in each species. Some groups (n = 1) considered alone have no scientific validity. also in the tables the data should be shown as mean ± SD

- How was the number of observations in each species decided? in some cases (n = 1 or n = 2) it seems to be a very small sample to draw conclusions.moreover in table 2 only two animals per experimental condition are shown.

- Line 376 the data for each animal could be reported in a table as supplementary material. in the results they should be shown as the group mean

- A graph for the groups treated with analgesics (for each species) could help in the presentation of the data

- Why was Friedmans test performed to evaluate differences in analgesic studies?
